# Insight into the Mechanism of Porcine Myofibrillar Protein Gel Properties Modulated by κ-Carrageenan

**DOI:** 10.3390/foods12071444

**Published:** 2023-03-29

**Authors:** Zhi Chen, Cheng Luo, Kangxu Wang, Yinji Chen, Xinbo Zhuang

**Affiliations:** 1College of Biological Science and Agriculture, Qiannan Normal University for Nationalities, Duyun 558000, China; chen2016@sgmtu.edu.cn; 2College of Food Science and Engineering, Nanjing University of Finance and Economics, Nanjing 210023, Chinachenyinji@nufe.edu.cn (Y.C.)

**Keywords:** fracture property, microstructure, protein gelation, κ-carrageenan

## Abstract

The purpose of this study is to explain the mechanism of porcine myofibrillar protein gel properties modulated by κ-carrageenan. The textural properties results showed that the stress at fracture of the composite gel with 0.4% κ-carrageenan had the highest value (91.33 g), which suggested that the 0.4% κ-carrageenan addition was the limitation. The strain at fracture was significantly reduced with κ-carrageenan addition. The composite gel with 0.4% κ-carrageenan had the lowest proportion of T22 (7.85%) and the shortest T21 relaxation time (252.81 ms). The paraffin section showed that the phase separation behavior of the composite gel transformed from single-phase behavior to dispersed phase behavior to bi-continuous phase behavior, and the ratio of CG/MP phase area significantly increased from 0.06 to 1.73. The SEM showed that the three-dimensional network of myofibrillar protein transformed from a loose structure to a compact structure to an unaggregated structure with κ-carrageenan addition. The myofibrillar protein network of the treatment with 0.4% κ-carrageenan had the highest DF value (1.7858) and lowest lacunary value (0.452). The principal component analysis was performed on the data of microstructure and textural properties, and the results showed that the dispersed phase behavior and moisture stabilization promoted the aggregation of myofibrillar protein and the composite gel had better water holding capacity and textural properties, while bi-continuous phase behavior hindered the aggregation of myofibrillar protein and the composite gel had worse water holding capacity and textural properties.

## 1. Introduction

With continuous economic development and health awareness improvement, consumers pay more attention to healthy diets and prefer foods with strict proportion control of fat, salt, or sugar [1,2]. Because of the juicy and elastic taste, processed meat products are popular with most consumers. However, the fat content in processed meat products is as high as 30%. Excessive intake of animal fat may cause a variety of chronic diseases such as atherosclerosis and elevated blood cholesterol [2,3]. Therefore, animal fat substitution is an inevitable trend in processed meat products. Relevant studies reported that polysaccharide-based fat substitutes simulated a similar taste to animal fat, and the sausages formulated with polysaccharide-based fat substitutes had acceptable sensory quality [1,4,5]. However, the regulatory mechanism of polysaccharides on microstructure and textural properties of myofibrillar protein needs further study.

Protein–polysaccharide gels are defined that the different nanoscale molecules (protein/polysaccharide) self-assemble and aggregate into micron-scale phase behavior structures [6]. The gel structures of low-fat processed meat products mainly include the skeleton structure (myofibrillar protein gel network) and the dispersed phase structure of polysaccharides. Myofibrillar protein (MP) has good gel capacity and determines the textural properties of final products. Gravelle, Marangoni, and Barbut [7] studied the relationship between the textural properties and microstructural characteristics of myofibrillar protein gelation. They found that the compactness of the myofibrillar protein network was positively correlated with the hardness and water holding capability. The MP–polysaccharide mixed system is thermodynamically incompatible at neutral pH, especially the thermal process intensifies the phase separation of the mixed system [8]. Van den Berg, Van Vliet, Van der Linden, Van Boekel, and Van de Velde [9] studied the influence of the volumes and spatial distribution of polysaccharides on the texture properties of soybean protein isolate gel through uniaxial compression test and confocal laser scanning microscopy. The results showed that polysaccharides as dispersed phases distributed uniformly in the protein gel network significantly improved the fracture stress of the composite gel system. Hence, the myofibrillar protein network and MP–polysaccharide phase separation behavior directly influenced the final textural properties and water holding capacity.

The κ-carrageenan (CG) is a water-soluble macromolecular polysaccharide obtained from edible red seaweeds. Numerous studies have reported that CG is a low-calorie dietary fiber and is widely used in commercial applications as gelling, thickening, and stabilizing agent, especially in processed meat products [10,11]. Totosaus, Alfaro-rodriguez, and Pérez-chabela [12] reduced the fat and sodium chloride content in sausages using κ-carrageenan and calcium chloride and found that κ-carrageenan incorporation improved the cooking yield and hardness of the low-fat products. Cao, Feng, Kong, Xia, Liu, Chen, Zhang, and Liu [13] studied the effect of various κ-carrageenan incorporation methods on the textural properties of frankfurters and found that the κ-carrageenan water suspension addition had the best effect on the improvement of the textural properties of frankfurters with satisfied sensory quality. The studies above focused on the κ-carrageenan utilization in sausage and its effect on the hardness and overall acceptance of sausage, but the underlying mechanism of this phenomenon had not been further explained from the molecular interaction between κ-carrageenan and myofibrillar protein. Zhang, Xu, Ji, Li, Wang, Xue, and Xue [14] studied the effect of κ-carrageenan at different ratios and heating temperatures on phase separation of the myofibrillar protein/κ-carrageenan mixing system. The temperature and κ-carrageenan concentration were positively correlated with the intensity of phase separation.

Hence, the objective of this study was to investigate the mechanism of porcine myofibrillar protein gel properties modulated by κ-carrageenan. Uniaxial compression tests were used to analyze the changes of textural properties. Low-field NMR was used to measure the moisture distribution in the composite gel network. Image analyses were used to analyze the microstructure of the MP-CG gel network. In addition, principal component analysis (PCA) was used to obtain a thorough comprehension of the relationship between the textural properties and microstructure.

## 2. Materials and Methods

### 2.1. Materials

The κ-carrageenan and fresh pork leg were purchased from Wanbang Industrial Co., Ltd. (Zhengzhou, China) and Sugo Supermarket, respectively.

### 2.2. Preparation of the Composite Protein System

The MP extraction was carried out as the method described by Luo, Zhang, Jiang, Chen, Zhou and Zhuang [15]. The ground muscle (about 200 g) was mixed with 800 mL extraction buffer (10 mM Na_2_HPO_4_/NaH_2_PO_4_, 0.1 mM NaCl, 2 mM MgCl_2_, 1 mM EGTA, pH 7.0, 4 °C), then the mixture was homogenized (T25, IKA, Inc., Königswinter, Germany) three times for 30 s at 6000 rpm/min. The homogenates were filtered through a 20-mesh sieve and centrifuged (Model 225, Beckman Coulter, Inc., Brea, CA, USA) at 3000× *g* for 15 min. The final pellet collected was pure MP. The extracted MP concentration was determined by the Biuret method and then diluted to 50 mg/mL (0.6 M NaCl, pH 7.0). The various concentrations of CG powder (0.0%, 0.05%, 0.1%, 0.2%, 0.4%, 0.8%, and 1.2%) were added to the MP solution and stirred evenly with a glass rod.

### 2.3. Uniaxial Compression Tests

All the MP–CG treatments were heated in the thermostatic water bath at 80 °C for 20 min. After cooling to 20 °C, the composite gel was cut into cylindrical pieces (20 mm in height, 20 mm in diameter). Samples were compressed to 75% of their initial height at speed of 1 mm/s through a texture analyzer (TA-XT plus, Stable Micro Systems Ltd., Godalming, UK) with the probe P/50. The sample deformation was expressed as strain at fracture, which was calculated as Equation (1) [16]:(1)$$\varepsilon_{H}=\int_{H_{0}}^{H}\frac{1}{H}dH=-\ln\left(\frac{H}{H_{0}}\right)$$
where *H*_0_ was the initial height and *H* was the compression height. The stress at fracture was calculated as Equation (2) [16]:(2)$$\sigma_{t}=\frac{F}{A}$$
where *F* was the compression force and *A* was the compressed area of the sample. The true stress accounts for the continuous change in the cross-sectional area assuming no change in a cylindrical shape and constant volume during the compression.

### 2.4. Dynamic Rheological Measurements

The MP–CG solution was loaded between two 50 mm diameter parallel plates with a 1 mm gap. The samples had the sweep strain (0.01–100%) at a frequency of 1 Hz to measure the linear viscoelastic region. The oscillatory test was conducted by applying a frequency of 1 Hz and the constant strain of 5% and heating from 20 °C to 80 °C with a heating rate of 2 °C/min.

### 2.5. Low-Field NMR

The MP–CG gels were placed into cylindrical glass tubes and measured through a Niumag Benchtop Pulsed NMR analyzer (Niumag PQ001; Niumag Electric Corporation, Shanghai, China) operating at 22.6 MHz. The T2 was measured using a Carr–Purcell–Meiboom–Gill (CPMG) with 32 scans, 12,000 echoes, 6.5 s between scans, and 250 μs between pulses of 90° and 180°. The MultiExp Inv Analysis software (Niumag Electric Corporation, Shanghai, China) was used to analyze the low-field NMR relaxation curves.

### 2.6. Microstructure Analysis

#### 2.6.1. The Phase Separation Behavior of Composite Gel System

The MP–CG gels (0.8 cm × 0.8 cm × 3 cm) were treated using formalin fixation, ethanol dehydration, and paraffin embedding procedures. The samples were cut into 8 μm thick sections by a microtome (CM1900, Leica, German), then stained with hematoxylin-eosin and finally observed through light microscopy (Axio Imager, Zeiss, Germany). ImageJ v1.47 software was used to calculate the ratio of the CG/MP phase area.

#### 2.6.2. Microstructure of Three-Dimensional Gel Network

The MP–CG gel was fixed with glutaraldehyde overnight. The MP–CG gel was treated using ethanol dehydration, freeze drying and gold spraying. Finally, it was observed through a Hitachi S3000 N scanning electron microscope (Tokyo, Japan) at an accelerating voltage of 20 kV. Micrographs were transformed into binary images and then used to calculate the fractal dimension and lacunary by ImageJ v1.47 software with FracLac 2.5v plugin.

### 2.7. Raman Spectroscopy of Protein Gels

The samples cut into slices (10 mm in height) were tested with an HR800 spectrometer (Horiba Jobi Yvon S.A.S., Longjumeau, France). The measurement parameters were set as follows: scan range 600–3050 cm^−1^; scans, 3; exposure time, 60 s; resolution, 2 cm^−1^. The spectra obtained were smoothed, baseline corrected, and normalized against the amplitude of the band at 1003 cm^−1^ (phenylalanine band) by Labspec version 5.0 (Horiba Jobi Yvon S.A.S., Longjumeau, France).

### 2.8. Statistical Analysis

All the measurements were performed at least in triplicates, and the data were expressed as mean ± SD. The data were analyzed using Statistical Analysis System 9.0 (SAS Institute Inc., Cary, NC, USA) with one-way ANOVA. Significant differences (*p* < 0.05) between means were identified using the least significance difference (LSD) procedure. Principal component analysis was performed with R language.

## 3. Results and Discussion

### 3.1. Uniaxial Compression Tests

The MP–CG composite gel is soft matter, and the typical textural properties include hardness or softness and elasticity or friability. The strain and stress at fracture were the major parameters used to reflect the hardness and elasticity of meat gelation. The stress and strain at fracture, respectively, have a positive correlation with the hardness and elasticity of the gelation [7,17,18]. The strain and stress at fracture of the composite protein gels with various CG addition were calculated by the equations and shown in Table 1. The results suggested that the CG addition significantly (*p* < 0.05) influenced the textural properties of the composite gel.

The stress at fracture of the MP–CG composite gel significantly improved with CG addition ranging in 0.05–0.4%, whereas CG addition beyond this range exhibited contrary results. With the CG addition reaching 0.8%, the stress at fracture of the MP–CG composite gel was even worse than the control. The stress at fracture of the MP–CG composite gel with 0.4% CG addition had the highest value of 91.33 KPa, indicating that 0.4% CG was the limitation in the MP–CG composite gel system. The related literature reported the same phenomenon that the polysaccharide addition had the limitation to improve the gel strength of final meat products [19,20]. The strain at fracture of the composite gel was significantly reduced with CG addition. The treatment with 1.2% CG addition had the lowest value of 0.62. The previous studies suggested that polysaccharides absorbed large volumes of moisture through hydrogen bonds and formed the hydrogel in the MP gel network [17,18]. The myofibrillar protein gel network was an elastic structure, while the CG hydrogel was low viscoelastic. The existence of CG hydrogel significantly reduced the strain at fracture of the composite gel, especially the 1.2% CG addition.

### 3.2. Dynamic Rheology

In Figure 1, the G′ of the composite gels was significantly affected by various CG addition during the linear heating process from 20 to 80 °C. During the thermal process, myofibrillar protein aggregated to form a gel network. The whole transformation was reflected by the curve G′. The curve G′ of pure MP mainly underwent three stages during the heating process: (1) the initial G′ rising from 45 to 50 °C: myosin head cross-linked to form dimerization body; (2) the further G′ reduction from 54 to 58 °C: myosin tail chain degraded and resulted in the collapse of the network structure; and (3) the final G′ rising above 60 °C: the exposed hydrophobic groups aggregated through hydrophobic interaction [21,22].

Compared to the control, the three-stage transition points of G′ did not change with CG addition (less than 0.4%), suggesting that MP was responsible for the continuous three-dimensional network and CG addition did not covalently cross-link with myofibrillar protein. Compared to the control, the final G′ of the composite gel with 0.4% CG addition significantly increased to 1191 Pa. The final G′ was positively correlated with firm aggregation and regular arrangement of the protein network [23]. The final rapid rising stage of G′ reflected that the exposed hydrophobic groups aggregated to form a three-dimensional network structure, which directly determined the hardness and elasticity of the composite gel. The CG addition below 0.4% significantly improved the aggregation of myofibrillar protein in the final stage. Jiang, Ma, Wang, Wang, and Zeng [11] found the same phenomenon that 0.5% κ-carrageenan addition significantly promoted the increase of G′ and indicated that the κ-carrageenan promoted the hydrogen bond interaction between oyster protein. However, the three-stage transition points of G′ significantly changed with the CG addition beyond 0.4%, suggesting that CG addition significantly hindered the aggregation of MP. The final G′ of the treatments with 1.2% CG addition significantly reduced to 566 Pa. Jiang, Ma, Wang, Wang, and Zeng [11] found that the G′ value of the treatments with the 2.0% pure κ-carrageenan addition was significantly lower than the control. The κ-carrageenan hydrogel was a brittle gel, and excess κ-carrageenan level caused the composite gel to be rigid or brittle.

### 3.3. NMR Proton Relaxation

The thermal process induced myofibrillar protein (MP) to form a porous spongy microstructure, while numerous moistures were immobilized in the three-dimensional gel network. The T2 relaxation time of moisture in the MP gel network was measured by the low-field NMR. Three distinct water categories mainly existed in the composite gel: (1) T2b, protein-associated water, relaxation time mainly between 1 and 10 ms; (2) T21, immobilized water trapped within the three-dimension network, relaxation time mainly between 200 and 400 ms; and (3) T22, free water, relaxation time mainly between 1000 and 2000 ms [24,25,26].

The specific relaxation times (T2b, T21, and T22) and corresponding proportions (PT2b, PT21, and PT22) of the composite gels with various CG addition are shown in Table 2. T2b relaxation time and its corresponding proportion of the treatments with various CG addition had no difference (*p* > 0.05). Numerous related studies had reported that polysaccharide addition did not affect the mobility and structural properties of protein-associated moisture [25,27]. The T21 relaxation time and its corresponding proportion reflected the water holding capacity of the composite gel and the yield of meat products. Compared to the control, the T21 relaxation time and its corresponding proportion of the treatment with 0.4% CG addition significantly decreased to 252.81 ms and increased to 88.66%, respectively. When CG addition reached 1.2%, the T21 relaxation time and its corresponding proportion of the treatment significantly increased to 401.40 ms and decreased to 72.09%, respectively. The compact gel structure strictly restricted moisture mobility and had a shorter T21 relaxation time. Hence, small CG addition promoted the aggregation of MP and the formation of a compact gel network, but excessive CG addition hindered the aggregation of MP and resulted in the formation of a loose gel network. The proportion of T22 of composite gel with 0.4% CG addition had the lowest value 7.85%. During the initial stage of heating, protein denaturation resulted in the exudation of the free water adsorbed by salt-soluble protein, and in the subsequent heating process, the aggregation of the protein network immobilized the free water. The heat-induced gel process of myofibrillar protein was the redistribution of water. The better the network structure was cross-linked, the less free water was. The treatment with 0.4% CG addition had the lowest proportion of T22, and it suggested that the treatment with 0.4% CG addition had the most compact MP gel network.

### 3.4. Microstructure of Composite Gel System

The phase separation behavior of the MP–CG gel and the microstructure of the MP gel network are shown in Figure 2. The red part is the MP gel network stained with hematoxylin–eosin, and the white circular part represents CG or moisture. Without CG addition, the MP gel network had a homogeneous structure, but some large volume and concentrated moisture cavities appeared in it. We further observed the MP gel network microstructure by using SEM. The three-dimensional network of pure MP gel had a loose cross-linked structure filled with numerous moisture channels. In salt solution, the MP depolymerized and adsorbed large volumes of moisture. The salt-soluble protein denatured and aggregated during the subsequent heating process, and the moisture adsorbed by MP resulted in moisture exudation. The moisture exudation happened in the gel network and formed moisture cavities or channels. The moisture channels hindered the aggregation of MP with the temperature further increasing. The imaging software was used to objectively and quantitatively expound the changes of phase separation behavior and the microstructure of the MP gel network with various CG additions. The changes of phase separation behavior were reflected through the ratio of CG/MP phase area. The aggregation of the MP gel network was reflected through the fractal dimension and lacunary. The values of fractal dimension and lacunary, respectively, had a positive and negative correlation with the aggregation of the MP gel network [28,29,30].

The paraffin section showed that the composite gel with CG addition presented two-phase separated structures. In the composite gel system, the MP gel network was the continuous phase and the CG was the dispersed phase, indicating that MP and CG had no direct interaction during the gelling process. Generally, the polysaccharide–protein mixture was a thermodynamically incompatible system, and the thermal process further promoted the formation of a two-phase separation gel system [14,31]. The paraffin section showed that the CG hydrogel formed numerous and small homogeneous cavities and trapped in the MP gel network of the treatment with <0.4% CG addition. Compared to the control, the composite gel network with the CG addition transformed from a homogeneous MP entirety into a two-phase separated structure. The ratio of the CG/MP phase area significantly increased from 0.065 to 0.719 with 0.4% CG addition in Table 3. We further observed the microstructure of the MP gel network with CG addition through SEM. The SEM micrograph showed that the CG addition at 0.05–0.4% promoted the formation of a dense and well-aggregated MP gel network. Compared to the control, the MP gel network with CG addition was a dense and compact structure, which suggested that MP was fully unfolded or sufficiently cross-linked with nearby MP during the heating treatment. In addition, the moisture channels distributed in the MP gel network were significantly reduced or even disappeared, and only small moisture pores were uniformly distributed in the MP gel network. Therefore, the actual function of CG addition was water immobilization and reduced random moisture exudation in the gel network.

However, the phase separation behavior of the composite gel transformed into bi-continuous structures with CG addition above 0.4%. In the paraffin section, CG formed continuous hydrogel through hydration and interpenetrated with the MP gel network. The CG hydrogel was no longer a dispersed phase. In the paraffin section, the ratio of CG/MP phase area significantly increased to 1.734 with 1.2% CG addition. The competitive hydration between protein and polysaccharide molecules significantly influenced the microstructure and textural properties of the composite gel system. The ratio of the CG/MP concertation was 0.08 and the ratio of CG/MP phase area was 0.719 in the composite gel with 0.4% CG addition, which reflected that the CG had higher hydration ability than MP. With the CG addition reaching 1.2%, the ratio of the CG/MP concertation was 0.24 and the ratio of CG/MP phase area was 1.734. These phenomena suggested that most moisture migrated to the CG in the mixture system. When the ratio of CG/MP phase area exceeded 1, it meant that the dominant structure in the composite gel transformed from the MP gel network to the CG hydrogel. The CG hydrogel had a similar negative influence as moisture channels, which severely hindered the aggregation of MP in the thermal process. Hence, the change of phase separation behavior resulted in an unaggregated MP gel network. The SEM micrographs showed that the three-dimensional MP gel network consisted of thick filaments, which were not fully unfolded or sufficiently cross-linked with nearby MP.

The phase separation behavior of the composite gel system with CG addition transformed from uniform single-phase behavior to dispersed phase separation behavior to bi-continuous phase separation behavior. However, hydration characteristics of CG could immobilize exudate moisture, vanish the appearance of moisture channels, and finally promote the aggregation of MP during the thermal process. Hence, the phase separation behavior and water immobilization resulted in the MP gel network transforming from a loose structure to a compact structure to an unaggregated structure.

### 3.5. Raman Spectral Analysis

The Raman spectra of the composite gel with various CG additions are shown in Figure 3. The frequency and intensity of the characteristic peaks of MP significantly were influenced by CG addition. The thermal process induced the unfolding of MP molecular conformation and the aggregation of hydrophobic groups. During the initial stage of heating, the double helix structure of the myosin tail denatured and unfolded. With the temperature further rising, exposed hydrophobic groups formed by the α-helix structure unfolding aggregated through hydrophobic interactions. In Raman spectra, the amide III (1645–1685 cm^−1^) region contained various characteristic peaks of secondary structure, which appeared in the range of 1658–1650, 1680–1665, and 1665–1660 cm^−1^, respectively. The characteristic peaks in these ranges, respectively, presented the α-helix structure, β-sheet structure, and random helix structure. Moreover, the correlation between the amide III band frequency and the protein structure conformation types was reported in many studies [32,33]. In addition, the intensity at 2945 cm^−1^ reflected the changes of aliphatic amino acids in MP, which had a negative correlation with the aggregation of hydrophobic groups.

The characteristic peak of the amide III band had a significantly right movement with CG addition, especially at 0.4% CG addition. It indicated that the CG addition promoted the unfolding of α-helix and the formation of β-sheet structure. Sánchez-González, Rodríguez-Casado, Careche, and Carmona [34] studied the effect of wheat dietary fiber on myofibrillar protein gel property. The results showed that the characteristic peak of the amide III band of treatment with dietary fiber shifted towards higher frequencies. In the present study, the results showed that the characteristic peak of the amide III band significantly shifted from 1670.2 to 1673.1 cm^−1^ with 0.4% CG addition. Xu, Han, Fei, and Zhou [35] studied the correlation between the textural properties of MP heat-induced gel and MP structure conformation. The results showed that the formation of β-sheets had a positive correlation with the aggregation of MP. Compared to the control, the characteristic peak of the amide III band had right movement from 1670.2 to 1668.3 cm^−1^ with CG addition at 1.2%. Hence, the exceeded CG addition hindered the unfolding of α-helix and the formation of β-sheet structure. The results of the characteristic peak in amide III were consistent with the unaggregated microstructure of the MP gel network.

The band intensity at 2945 cm^−1^ reflected the changes of aliphatic amino acids in peptides and proteins. The intensity at 2945 cm^−1^ of treatments with various CG addition is shown in Table 4. The band intensity at 2945 cm^−1^ of treatment with 0.4% CG addition significantly reduced from 5.31 to 4.33. Additionally, it suggested that CG addition promoted the aggregation of hydrophobic groups. However, the band intensity at 2945 cm^−1^ of composite gel with 1.2% CG significantly increased to 6.06, indicating that the exceeded CG addition hindered the aggregation of hydrophobic groups.

### 3.6. Principal Component Analysis

The related data of the MP–CG gel are presented as a principal component analysis (PCA) in 4. The first principal component contributed to 73.4% of the total variation and the second principal component contributed to 84.3% of the total variation. It indicated that the physicochemical characteristics (stress and strain) and microstructure (fractal dimension, lacunary value, CG/MP phase ratio, and molecular conformation) were highly correlated. The loading plot (Figure 4A) showed that the first principal component was positively related with fractal dimension, final G′, stress at fracture, and Amide III and was negatively related with lacunary value, the band intensity of 2945 cm^−1^, and T21. It indicated that the T2 relaxation time and the textural property were modulated by MP molecular conformation (Amide III and the band intensity of 2945 cm^−1^) and the aggregation of the MP three-dimensional network (fractal dimension and lacunary value). Moreover, the second principal component was positively related to CG/MP phase ratio and was negatively related to strain at fracture. It reflected that the strain at fracture was just simply modulated by CG/MP phase ratio.

The principal component analysis (Figure 4B) grouped the composite MP gel with various CG additions into seven well-differentiated clusters, indicating that polysaccharide addition had a significant influence on the physicochemical properties, phase separation behavior, and the MP three-dimensional network. In Figure 3B, the first principal component was *X*-axis, and the second principal component was *Y*-axis. The groups (0.2% and 0.4%) in the first quadrant represented high hardness and low elasticity gel system; the groups (0.8% and 1.2%) in the fourth quadrant represented low hardness and low elasticity gel system; and the groups (the control and 0.05%) represented moderate hardness and high elasticity gel system. The results indicated that the CG addition had a positive effect on the hardness of the MP–CG gel, but it had a limitation at 0.4%. The composite gel with 0.8 and 1.2% CG addition was distributed in the negative direction of the first principal component, and it suggested the degradation of the MP gel network. However, the composite gel with CG addition migrated in the positive direction of the second principal component, indicating that the CG addition would reduce the elasticity.

### 3.7. Mechanistic Explanation

The mechanistic explanation of porcine myofibrillar protein gel property modulated by κ-carrageenan is exhibited in Figure 5. The three-dimensional network of treatments with various CG additions transformed from a loose structure to a compact structure to an unaggregated structure. The thermal processing induced the denaturation of MP absorbed moisture. Additionally, moisture exudation formed moisture channels and was irregularly/randomly distributed in the MP gel network. The irregular moisture channels hindered the aggregation of hydrophobic groups, leading to a loose three-dimensional gel network. With CG addition, the CG absorbed moisture through hydration bonds and homogeneously dispersed in MP solution. During the heating process, the compact three-dimensional gel network was formed without moisture channels. The compact three-dimension gel network in turn firmly immobilized the CG hydrogel, and it reduced T21 relaxation time. Moreover, the stress at fracture and final G′ of the composite gel also improved.

However, the CG hydrogel interpenetrated with the MP gel network with the CG addition reaching 0.8%. The phase separation behavior of the MP–CG gel system transformed from the dispersed phase separation behavior to bi-continuous phase separation behavior. The CG hydrogel as moisture channels had a negative influence on the aggregation of the MP during the thermal process. Hence, the change of phase separation behavior resulted in the fragmentation of the gel network, leading to an unaggregated gel network. The molecular conformation analysis also showed that the exceeded CG addition hindered the unfolding of α-helix and the aggregation of hydrophobic groups. The bi-continuous phase separation structure seriously reduced the WHC and facture stress of the composite gel system.

CG could immobilize exudate moisture, limit the moisture channels distributed in the gel network, and finally promote the aggregation of MP. However, the phase separation behavior of MP–CG composite gel was also significantly influenced by the CG addition. The phase separation behavior of MP–CG composite gel transformed from single-phase behavior to dispersed phase behavior to bi-continuous phase behavior. Hence, the phase separation behavior and water immobilization modulated the aggregation of the MP gel network, further influencing T2 relaxation time and textural properties. Moreover, the myofibrillar protein gel network was elastic, while the CG hydrogel was low viscoelastic. The existence of the large CG cavities significantly reduced the fracture strain of the composite gel with CG addition.

## 4. Conclusions

With various CG addition, the MP three-dimensional network transformed from a loose structure to a compact structure to an unaggregated structure. The hydration property of CG immobilized moisture and CG was homogeneously trapped in the MP solution as the dispersed phase. The elimination of moisture channels promoted MP aggregation and formed a compact three-dimensional network. However, the CG hydrogel and MP gel network formed a bi-continuous phase separation behavior with CG addition above 0.4%. The CG hydrogel as moisture channels hindered the MP aggregation, and an unaggregated MP three-dimensional network was formed. The compactness of the MP gel network had a positive correlation with the fracture property and water holding capacity. Hence, the CG addition below 0.4% significantly improved the physicochemical characteristics of the MP–CG composite gel, which was important to provide a design thought for CG utilization in processed meat products and the development of low-fat meat products with good sensory texture.

## Figures and Tables

**Figure 1 foods-12-01444-f001:**
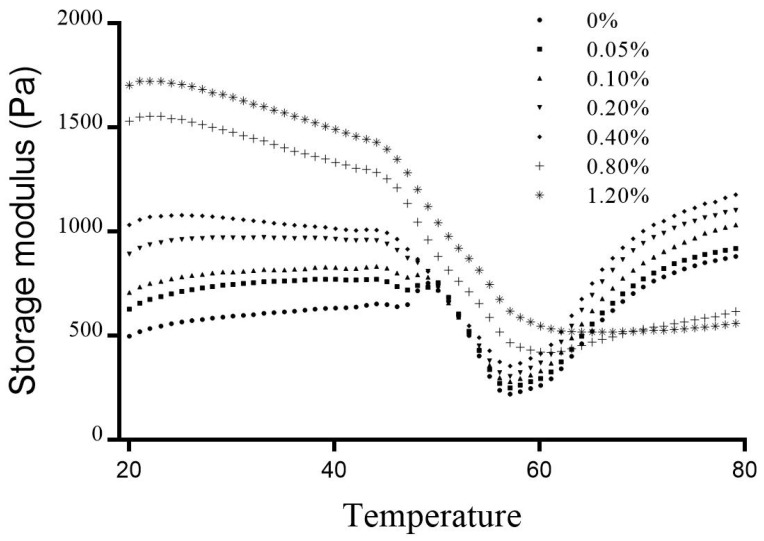
Storage modulus (G′) of the composite gels with various CG addition. 0%: the control; 0.05%: MP with 0.05% CG; 0.10%: MP with 0.1% CG; 0.20%: MP with 0.2% CG; 0.40%: MP with 0.4% CG; 0.80%: MP with 0.8% CG; 1.20%: MP with 1.2% CG.

**Figure 2 foods-12-01444-f002:**
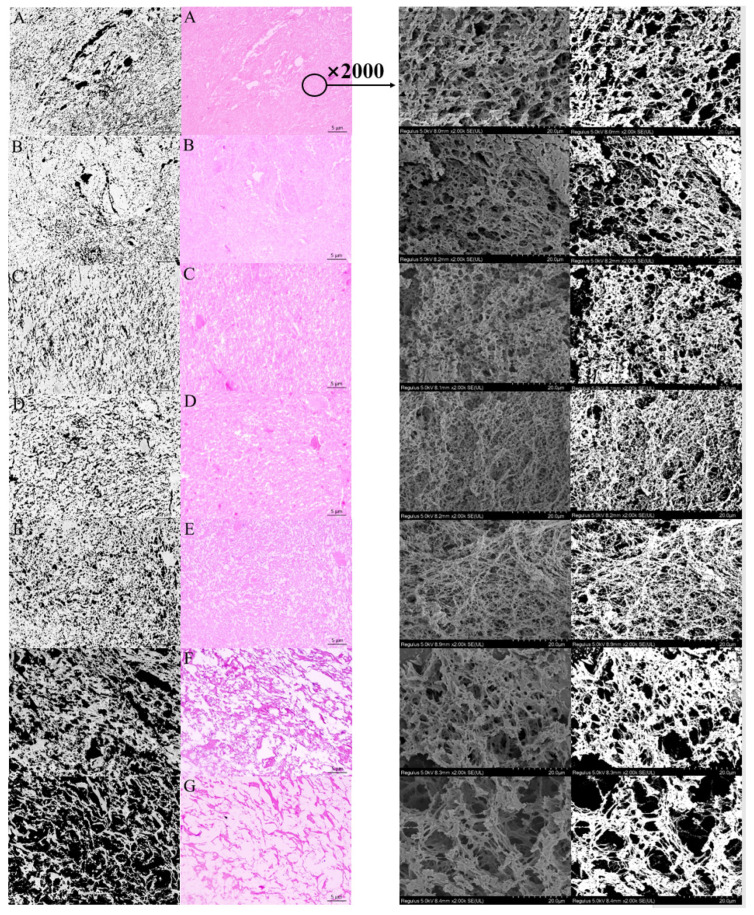
Micrograph images (400×) and the corresponding threshold binary images of the composite gels (**left**); SEM images (2000×) and the corresponding threshold binary images of the MP gel network (**right**). (**A**): the control; (**B**): MP with 0.05% CG; (**C**): MP with 0.1% CG; (**D**): MP with 0.2% CG; (**E**): MP with 0.4% CG; (**F**): MP with 0.8% CG; (**G**): MP with 1.2% CG.

**Figure 3 foods-12-01444-f003:**
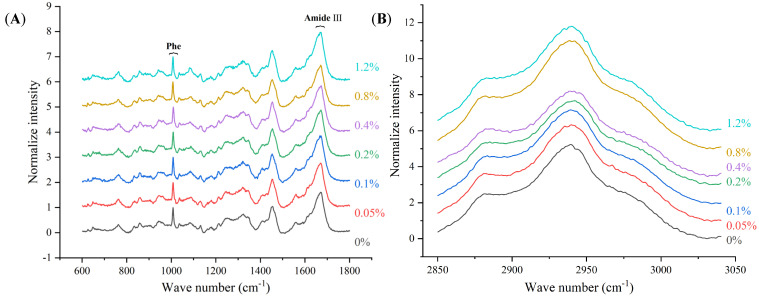
Raman spectrum between 600 and 1800 cm^−1^ of composite MP gels concentrations of CG (**A**); Raman spectrum between 2850 and 3050 cm^−1^ of composite MP gels concentrations of CG (**B**). 0%: the control; 0.05%: MP with 0.05% CG; 0.10%: MP with 0.1% CG; 0.20%: MP with 0.2% CG; 0.40%: MP with 0.4% CG; 0.80%: MP with 0.8% CG; 1.20%: MP with 1.2% CG.

**Figure 4 foods-12-01444-f004:**
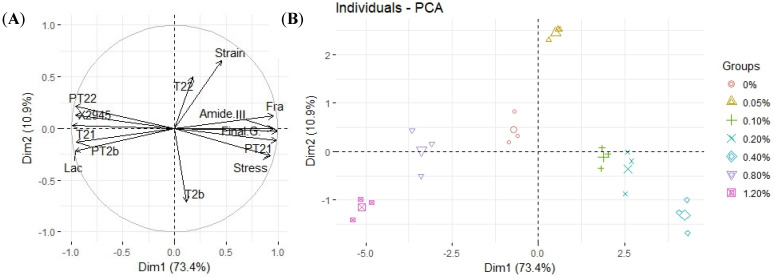
Principal component (PC) analysis loading plots in the plane of PC2 vs. PC1 (**A**). Results of the PCA score plots in the PC1 vs. PC2 plane (**B**).

**Figure 5 foods-12-01444-f005:**
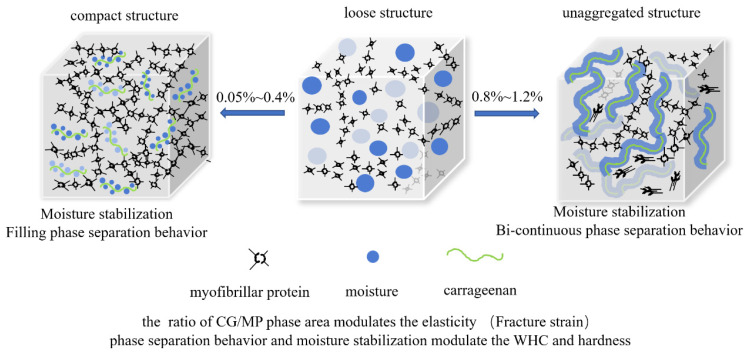
The physicochemical characteristics of MP gels influenced by CG through phase behavior and water immobilization.

**Table 1 foods-12-01444-t001:** The stress and strain at fracture of the MP composite gels with various CG.

Treatments	Stress a at Fracture (KPa)	Strain at Fracture
0%	56.65 ± 1.55 e	0.95 ± 0.01 a
0.05%	63.05 ± 1.91 d	0.89 ± 0.03 b
0.1%	73.24 ± 1.69 c	0.87 ± 0.02 b
0.2%	79.97 ± 1.85 b	0.77 ± 0.02 c
0.4%	91.22 ± 1.61 a	0.73 ± 0.03 c
0.8%	51.41 ± 0.87 f	0.65 ± 0.02 d
1.2%	49.34 ± 2.05 f	0.62 ± 0.03 d

0%: the control; 0.05%: MP with 0.05% CG; 0.10%: MP with 0.1% CG; 0.20%: MP with 0.2% CG; 0.40%: MP with 0.4% CG; 0.80%: MP with 0.8% CG; 1.20%: MP with 1.2% CG. Different letters (a–f) in a column indicate significant differences (*p* < 0.05) between treatments.

**Table 2 foods-12-01444-t002:** Relaxation times and corresponding peak areas of composite MP gels with various CG.

Treatments	T2b (ms)	PT2b (%)	T21 (ms)	PT21 (%)	T22 (ms)	PT22 (%)
0%	4.79 ± 0.55 abc	3.69 ± 1.26	350.61 ± 1.38 b	78.45 ± 2.13 d	2724 ± 539	17.65 ± 2.14 b
0.05%	4.25 ± 1.61 d	3.76 ± 1.47	312.71 ± 3.99 c	81.09 ± 2.65 d	2893 ± 484	15.61 ± 0.69 b
0.1%	4.67 ± 1.67 a	3.69 ± 1.11	292.20 ± 5.02 d	84.19 ± 2.21 c	2777 ± 550	11.95 ± 1.96 c
0.2%	4.31 ± 0.85 cd	3.75 ± 1.28	271.17 ± 5.70 e	86.79 ± 0.58 b	2669 ± 574	9.77 ± 0.75 d
0.4%	4.89 ± 1.61 ab	4.09 ± 1.71	252.81 ± 3.75 f	88.66 ± 0.41 a	2862 ± 311	7.85 ± 0.62 e
0.8%	5.41 ± 0.87 bcd	4.03 ± 1.75	389.30 ± 3.66 a	75.87 ± 2.53 e	2674 ± 469	20.01 ± 2.84 a
1.2%	5.17 ± 0.29 abc	4.40 ± 1.72	401.4 ± 14.72 a	72.09 ± 2.79 e	2760 ± 563	23.37 ± 1.32 a

Different letters (a–f) in a column indicate significant differences (*p* < 0.05) between treatments.

**Table 3 foods-12-01444-t003:** The ratio of CG/MP phase area, DF, and lacunary of composite MP gels with various CG.

Treatments	CG/MP Ratio	DF	Lacunary
0%	0.06 ± 0.04 f	1.8572 ± 0.0025 c	0.483 ± 0.010 c
0.05%	0.12 ± 0.04 ef	1.8601 ± 0.0022 c	0.466 ± 0.005 d
0.1%	0.20 ± 0.02 e	1.8687 ± 0.0030 b	0.463 ± 0.003 de
0.2%	0.34 ± 0.02 d	1.8747 ± 0.0017 a	0.456 ± 0.003 ef
0.4%	0.72 ± 0.05 c	1.8757 ± 0.0016 a	0.452 ± 0.005 f
0.8%	1.13 ± 0.07 b	1.8411 ± 0.0037 d	0.511 ± 0.010 b
1.2%	1.73 ± 0.10 a	1.8176 ± 0.0047 e	0.543 ± 0.006 a

Different letters (a–f) in a column indicate significant differences (*p* < 0.05) between treatments.

**Table 4 foods-12-01444-t004:** The frequency of characteristic peak in Admin III and normalized intensities of 2945 cm^−1^ band of composite MP gels with various CG.

Treatments	Admin III (cm^−1^)	I2945/1003
0%	1670.2 ± 0.94 d	5.31 ± 0.11 c
0.05%	1670.8 ± 0.54 cd	5.24 ± 0.05 c
0.1%	1671.8 ± 0.54 bc	5.09 ± 0.10 c
0.2%	1672.4 ± 0.94 ab	4.57 ± 0.15 d
0.4%	1673.1 ± 0.94 a	4.33 ± 0.22 e
0.8%	1668.3 ± 0.94 e	5.74 ± 0.15 b
1.2%	1667.69 ± 0.53 e	6.06 ± 0.16 a

Different letters (a–e) in a column indicate significant differences (*p* < 0.05) between treatments.

## Data Availability

The data will be available if required.

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
