# Peer review of "Insight into the Mechanism of Porcine Myofibrillar Protein Gel Properties Modulated by κ-Carrageenan"

_foods, 2023, doi:10.3390/foods12071444_

Round 1

Reviewer 1 Report

Comments and suggestions to authors are described in the attached document.

Reviewer 2 Report

Dear Authors,

This manuscript aims to contribute to the explanation of the mechanism of porcine myofibrillar protein gel property, modulated by the polysaccharide κ-carrageenan (CG). It comprises the abstract, keywords, introduction, material and methods, results and discussion, conclusions, 39 references, 6 figures and 2 tables.

However, the structure and some content of the manuscript seem confusing, as:

1) The numbering of some sections/subtitles is repeated (“3.5 NMR proton relaxation” – line 176 vs. “3.5 Mechanistic explanation” - line 316) and is out of order of appearance (“3.3.1 Microstructure of composite gel system” – line 197 vs. “3.1 Raman spectral analysis – line 253”); maybe 3.3.1 should be 3.3…;

2) Two sections/subtitles seem to be missing (section 2.1 before line 70; section 3.1 before line 129);

3) Some sentences seem incomplete (line 133 “The results suggested that the CG addition significantly (P<0.05) influenced the composite gel”; which characteristic of the composite gel was influenced by the CG addition?) and there are gross errors (line 181, “40” instead of “400”; line 233 “CG/MP concertation” instead of “CG/MP concentration”), possibly due to language differences;

4) Several in-text citations should be presented only by [numbers] (examples in lines: 42, 45, 53, 55, 58, 60, 73, 84, 95, 101, 113, 121, 271, 274). Please harmonize the presentation of units after each value (with vs without space).

So, a sorrow revision of the document is recommended.

Please see below for specific comments:

1. Introduction

Line 51: Please replace “κ-carrageenan (CG) is” by “The κ-carrageenan (CG) is”.

2. Materials and Methods

Line 123: Please include the reference number for “(Herrero, 2008), add in the list of references and renumber other subsequent references, if applicable.

    3. Results and discussion   The first paragraph of section 3.5 (lines 177 – 182) does not seem to contain results; please consider changing this text elsewhere (include in material and methods?)   Line 181: Please replace “relaxation time mainly between 200 to 40ms” by “relaxation time mainly between 200 to 400ms”   Line 233: Please replace “CG/MP concertation” by “CG/MP concentration”.     Figures

The word “Figure” should de used instead of “Fig” for legends of figures 1, 3 – 6. Also, please revise the legend of figures 1, 2 and 4, as no letters A to G appear in these figures.

    Best Regards,  

Reviewer

Reviewer 3 Report

An interesting paper that brings a lot of discussion into the mechanisms of the use of K-carrageenan.

See comments below:

Line 29.  Can remove the definite article "the".

Line 31-32.  If intake of animal fat is such a problem, perhaps better references should be used.  I would suggest words like "may" be more appropriate.

Line 35.  This is awkwardly written.  Please revise.

Line 63.  Consider making "property? plural.

Line 127.  Authors should use something other than Duncan's multiple range test.  The error rates may not be what the authors intended.

Line 138.  This is awkwardly written.  Please revise.

Line 193.  Consider formatting the foot notes to the tables to be the same width as the table itself.  Also, the authors should consider adding a superscript to the footnotes.

Line 200.  Is the use of the word "humongous" getting at the idea the authors want to convey here?  Is it the size that is important or something about cavities observed?

Line 203.  Is "lager" spelled correctly?  Is it supposed to be "larger"?

Line 312.  "nrtwork".  Is this supposed to be "network"?

Line 217.  Consider adding "a" between "was" and "thermodynamically."

Line 224.  Consider a different work than "Besides".  Perhaps, "In addition. . ." or similar.

Line 230.  Consider making "were" singular.

Line 246 & 267.  Please include superscripts for the footnote and make the footnote the same width as the table.

Line 289.  Although the footnotes show the different (A) and (B) graphs, the graphs themselves are not labeled. 

Line 308.  Not sure what the "additive limitation" is.

Line 314.  Although the footnotes show the different (A) and (B) graphs, the graphs themselves are not labeled.

Line 334.  This is not a complete sentence.  Please revise.
